

# Association between celiac disease and fibromyalgia and their severity: a cross-sectional study

Mehmet Serkan Kılıçoğlu[1], Safiye Sayılır[1], Ozan Volkan Yurdakul[1],
Teoman Aydin[1], Koray Koçhan[2], Metin Basaranoglu[2] and
Okan Kucukakkas[3]

[1] Department of Physical Medicine and Rehabilitation, Bezmialem Vakif University, Istanbul, Fatih, Turkey
[2] Department of Gastroenterology, Bezmialem Vakif University, Istanbul, Fatih, Turkey
[3] Department of Physical Medicine and Rehabilitation, NMC Royal Hospital, Khalifa City, Abu Dhabi, United Arab Emirates

## ABSTRACT

**Background:** Fibromyalgia (FMS) is a common musculoskeletal disorder with many causes. People with fibromyalgia often have the same symptoms as people with celiac disease (CD). Demonstration of the coordination and frequency of FMS and CD is important for effective treatment.

**Methods:** This is a single center cross-sectional clinical study. The study included 60 patients who were diagnosed with CD by the Gastroenterology Clinic based on American College of Gastroenterology (ACG) criteria. Patients were also asked to complete the Widespread Pain Index (WPI), Symptom Severity Scale (SSS), and Fibromyalgia Impact Questionnaire (FIQ) to diagnose fibromyalgia and assess its severity. The results were used to analyze the frequency of concomitance and relationship between the two diseases.

**Results:** The relationship between the clinical types of CD and the presence of fibromyalgia was insignificant. Analysis of the relationship between the pathologic typing of biopsy and fibromyalgia frequency was insignificant. Those with antibodies more frequently met criteria for fibromyalgia ($P = 0.04$, $P = 0.04$, respectively).

**Conclusions:** Presence of clinical extraintestinal manifestations in patients with CD should lead clinicians to consider FMS as a possible diagnosis. This points to the importance for clinicians in all subspecialties to be aware of the various symptoms and diseases associated with FMS.

# INTRODUCTION

Celiac disease (CD) is an autoimmune gluten-dependent enteropathy characterized by small intestinal atrophy in susceptible individuals (*Catassi et al., 2022*). Its prevalence in European populations has been reported to be approximately 1%. Diarrhea is the most common symptom in children with the typical form of the disease. In adults, it is much rarer than in children and occurs as an initial symptom in approximately 50% of patients. The disease may manifest with a combination of gastrointestinal and extraintestinal

Corresponding author
Mehmet Serkan Kılıçoğlu,
dr.serkan.kilicoglu@gmail.com

symptoms such as anemia, osteoporosis, dermatitis herpetiformis, diabetes mellitus, growth failure, and nonspecific hypertransaminasemia or it may manifest with extraintestinal symptoms alone without typical manifestations (*Lindfors et al., 2019*).

Fibromyalgia syndrome (FMS) is a common cause of chronic musculoskeletal pain and has a prevalence of 2–5% (*Queiroz, 2013*). The most common and characteristic symptoms of FMS include widespread pain, morning stiffness, waking unrefreshed, and sleep disturbance. Gastrointestinal manifestations, which are common in patients with fibromyalgia, have been associated with irritable bowel syndrome due to common comorbidities in 32% to 81% of patients (*Veale et al., 1991*; *Sperber et al., 1999*; *Kurland et al., 2006*; *Bayrak, 2020*; *Yepez et al., 2022*). In terms of intestinal symptoms, CD and fibromyalgia may present similar manifestations. CD and FMS share many clinical symptoms such as abdominal pain, bloating, diarrhea/constipation, fatigue, widespread musculoskeletal pain, depression, and other mood/anxiety disorders. This similarity suggests that CD may be the root of symptoms in at least a subgroup of patients with fibromyalgia (*García-Leiva et al., 2015*). Previous studies have reported an association between FMS and autoimmune diseases such as rheumatoid arthritis, systemic lupus erythematosus, and Hashimoto's thyroiditis, but the association between fibromyalgia and CD has not been adequately investigated (*García-Leiva et al., 2015*; *Weir et al., 2006*).

Many of the gastrointestinal symptoms seen in fibromyalgia, such as abdominal pain, bloating, diarrhea, or constipation, are characteristic of people with CD or gluten sensitivity. In contrast, gastrointestinal symptoms such as joint and/or muscle pain, chronic fatigue, or changes in elevated values have been observed in patients with CD or gluten sensitivity in fibromyalgia patients.

The present study aimed to examine clinical manifestations in adult patients with CD, to investigate the frequency and severity of fibromyalgia that co-occurs with CD, and to analyze the possible relationship between the two diseases. Thus, the study seeks to provide a clearer insight into the co-occurrence of fibromyalgia and CD to contribute to early diagnosis and treatment.

## MATERIALS AND METHODS

### Study model and setting

This is a single center, cross-sectional clinical study. Patients were asked to complete a demographic questionnaire prepared by the researchers that inquired about patient characteristics such as age, sex, occupation, medications, and comorbidities. Patients were classified into clinical types of CD—1, Classic Disease; 2, Non-classic Disease; 3, Silent Disease; 4, Potent Disease (*Ferguson, Arranz & O'Mahony, 1993*). For patients who underwent biopsy and their results were available, they were also classified using modified Marsh classification. Further, patients were asked to complete the Widespread Pain Index (WPI), Symptom Severity Scale (SS scale), and Fibromyalgia Impact Questionnaire (FIQ) to diagnose fibromyalgia and assess its severity. The results were used to analyze the frequency of concomitance and relationship between the two diseases.

## Population

The study included patients who were diagnosed with CD by the Gastroenterology Department according to the American College of Gastroenterology (ACG) criteria (*Rubio-Tapia et al., 2013*) and who were referred to the corresponding author's Physical Medicine and Rehabilitation Department between September 2022 and December 2022. Five patients were excluded for not signing the consent form, and four patients were excluded for providing incomplete questionnaires. Inclusion criteria: patients over 18 years of age agreed to participate in the study, signed the consent form, were able to complete the questionnaires and measurements and were followed up for CD. Exclusion criteria: Patients with an unconfirmed diagnosis of CD, patients who refused to participate in the study, and patients with insufficient cognitive faculty.

Patients were informed by the attending physiatrist about the purpose of the study, and those who were interested and willing to participate were enrolled after signing a consent form.

## Ethical aspects

Only participants who can and are willing to provide written informed consent were eligible to participate in the current clinical trial. This study received IRB approval from the Bezmialem Foundation University Ethics Committee (Decision Number: 2019-3546) and was in accordance with the Declaration of Helsinki and the International Conference on Harmonization Principles.

## Duodenal biopsy analysis

All patients in this study underwent gastroduodenoscopy and at least four duodenal biopsies in gastroenterology service. The diagnosis of celiac disease is established when duodenal biopsy samples showing increased intraepithelial lymphocytes with crypt hyperplasia (Marsh type 2), or, more commonly, also with villous atrophy (Marsh type 3) in a patient with positive celiac serology (*Aziz et al., 2010*; *De Leo et al., 2015*). Samples were routinely stained with hematoxylin and eosin and anti-CD3 immunohistochemical monoclonal antibodies to confirm and quantify the presence of intraepithelial lymphocytes (IELs). These samples were examined by two participating physicians and classified into the following categories: Stage 0: histologically normal duodenum; Stage 1: increased IEL infiltration with ≥25% total epithelial cells; Stage 2: Crypt hyperplasia and diffuse inflammatory infiltration. In lamina propria; Stage 3: villous atrophy is divided into the following groups defined by Marsh in 1992: 0, normal; 1, increased IEL; 2, increased IELs and crypt hyperplasia; 3a, partial villous atrophy; 3b, total villous atrophy; 3c, complete villous atrophy (*Marsh, 1992*; *Bodd et al., 2012*).

## Widespread pain index and symptom severity scale

The WPI measures the presence of pain in 19 areas of the body (including the neck, right arm, and left arm) over the past 7 days (*Wolfe et al., 2010*; *Clauw, 2014*; *Yanmaz, Atar & Biçer, 2016*). Point for each task is equal to 1. Individual items are summed to create a total score; higher scores indicate more severe pain. The 6-item SSS measures (1) the presence of

symptoms (abdominal pain, headache, depression, *etc.*) in the last 6 months and (2) the severity of cognitive symptoms (fatigue, difficulty thinking or remembering, tiredness in the morning) in the last 7 days. Individuals were asked whether these symptoms were related to pain in general, specifically, or were caused by pain. The presence of symptoms equals 1 point. The severity of cognitive symptoms is measured on a 4-point scale; where 0 represents "no problem" and three represents "serious problem." Scores were calculated by summing the items; higher scores (scores above 12) indicated more severe symptoms. WPI and SS scale scores can be combined to create a total score (range 0–31); higher scores indicate more pain characteristic. This measure includes two additional questions that do not contribute to the total score; the first measures the duration of symptoms, the second determines whether the symptoms are caused by the condition in the first place.

All participants completed the WPI and SS scales. The authors have permission to use this instrument from the copyright holders.

### Fibromyalgia impact questionnaire

FIQ is composed of 10 items. The first item includes 11 Likert-type questions. These questions are rated on a scale from 0 to 3 and averaged. The second item inquiries about the number of days the patient felt good in the last week. The answer of this item is reverse-scored (0 days = 7 and 7 days = 0). The third item asks about the number of days the patient was unable to work in the last week. Scores of the first three items are normalized. The score obtained from the first item is multiplied by 3.3, and the score obtained from the second and third items is multiplied by 1.4. The other seven items include questions assessing the severity of symptoms, pain, fatigue, waking unrefreshed, stiffness, anxiety, and depression using a 10-point Visual Analog Scale. Answers to these questions yield a score between 0 and 100 (*Sarmer, Ergin & Yavuzer, 2000*). The average score for a patient with fibromyalgia is 50, and a higher score indicates greater physical disability. In the present study, all participants completed the FIQ. The authors have permission to use this instrument from the copyright holders.

### Statistical analysis

Statistical data were obtained for continuous variables (mean, standard deviation, and range) and standardized variables (percentages). Normally distributed results from measurements were analyzed using Student's t test or one-way ANOVA followed by *post hoc* Fisher's test as appropriate. Categorical data were analyzed using the chi-square probability test (or Fisher's exact test when relevant). All statistical analyzes were performed using SPSS 15.0 (SPSS Inc., Chicago, IL). Statistical significance was set at $P < 0.05$.

## RESULTS

Demographic and clinical parameters relating to 60 patients with CD included in the study are shown in Table 1. Of the patients, 50 were female (83%). Of the 17 patients diagnosed with FMS, 16 (94%) were female. The mean duration of CD was 78.75 months.

**Table 1 Demographic and clinical parameters.** Clinical and demographic data of celiac patients included in the study. There is data collected about co-morbidities and antibody types.

| Demographic and clinical parameters | Mean ± SD/$n$ (%) |
|---|---|
| Gender, $n$ (%) | |
| Male | 10 (17) |
| Female | 50 (83) |
| Age | 39.85 ± 11.18 |
| Weight (kilogram) | 64.55 ± 13.47 |
| Height (meter) | 162.77 ± 7.67 |
| BMI (kg/m$^2$) | 24.38 ± 5.01 |
| Duration of disease (Month) | 78.75 ± 79.86 |
| Education level | |
| Primary school | 30 (50) |
| Secondary school | 8 (14) |
| High school | 11 (18) |
| University | 11 (18) |
| Profession | |
| Worker | 23 (38) |
| Student | 3 (5) |
| Unemployed | 33 (55) |
| Retired | 1 (2) |
| Hypertension | |
| Present | 6 (10) |
| Absent | 54 (90) |
| Diabetes mellitus | |
| Present | 8 (13) |
| Absent | 52 (87) |
| Asthma | |
| Present | 6 (10) |
| Absent | 54 (90) |
| Thyroid diseases | |
| Present | 16 (27) |
| Absent | 44 (73) |
| Rheumatic diseases | |
| Present | 3 (5) |
| Absent | 57 (95) |
| Fibromyalgia | |
| Present | 17 (28) |
| Absent | 43 (72) |
| Widespread pain index | 6.37 ± 5.43 |
| Symptom severity scale | 4.60 ± 2.64 |
| Fibromyalgia impact questionnaire | 42.15 ± 21.54 |
| Celiac disease clinical classification | |
| Unclassified | 1 (2) |

(Continued)

| Table 1 (continued) | |
|---|---|
| **Demographic and clinical parameters** | **Mean ± SD/$n$ (%)** |
| Classic | 43 (72) |
| Nonclassic | 4 (7) |
| Potential | 1 (2) |
| Silent | 11 (19) |
| Pathological type of biopsy | |
| Not classified | 13 (22) |
| Type 1 | 12 (20) |
| Type 2 | 13 (22) |
| Type 3 | 21 (34) |
| Type 4 | 1 (2) |
| Anti-gliadin antibodies | |
| Not analyzed | 13 (21) |
| Negative | 28 (47) |
| Positive | 19 (32) |
| Antireticulin antibody | |
| Not analyzed | 55 (92) |
| Negative | 4 (7) |
| Positive | 1 (1) |
| Tissue transglutaminase antibody | |
| Not analyzed | 1 (2) |
| Negative | 18 (30) |
| Positive | 41 (68) |
| Endomysial antibodies | |
| Not analyzed | 6 (10) |
| Negative | 28 (47) |
| Positive | 26 (43) |

The majority of patients in this study were diagnosed with transglutaminase antibodies (41 patients) and endomysium antibodies (26 patients). Of the patients with CD included in the study, 17 (28%) were diagnosed with fibromyalgia.

The relationship between the clinical types of CD and the presence of fibromyalgia was insignificant ($P = 0.11$). Analysis of the relationship between the pathologic typing of biopsy and fibromyalgia frequency was insignificant ($P = 0.88$). On the other hand, analysis of the relationship between antibodies and the frequency of fibromyalgia showed a lower frequency of fibromyalgia in those who were negative for tissue transglutaminase antibodies and endomysium antibodies compared with those who were positive for these antibodies ($P = 0.04$ and $P = 0.04$, respectively; Table 2).

There was a positive relationship between FIQ and type of CD treatment, type of biopsy, and reaction in fibromyalgia patients (Table 3). Further, there was no significant relationship between the WPI, SS scale, and CD parameters.

**Table 2 Fibromyalgia and celiac disease parameters.** The relationship between antibody values of celiac patients with fibromyalgia. Bold values indicate critical statistically significant outcomes.

| Fibromyalgia | | Absent | Present | P |
|---|---|---|---|---|
| Clinical type of CD | Classic | 32 (74%) | 11 (26%) | 0.109 |
| | Nonclassic | 4 (100%) | 0 (0%) | |
| | Silent | 5 (46%) | 6 (54%) | |
| | Potential | 1 (100%) | 0 (0%) | |
| Pathological type of biopsy | Type 1 | 8 (67%) | 4 (33%) | 0.875 |
| | Type 2 | 10 (77%) | 3 (23%) | |
| | Type 3 | 16 (76%) | 5 (24%) | |
| | Type 4 | 1 (100.00%) | 0 (0.00%) | |
| Anti-gliadin antibodies | Negative | 16 (57%) | 12 (43%) | 0.217 |
| | Positive | 15 (79%) | 4 (21%) | |
| Endomysium antibodies | Negative | 16 (57%) | 12 (43%) | **0.04** |
| | Positive | 22 (85%) | 4 (15%) | |
| Anti-reticulin antibodies | Negative | 2 (50%) | 2 (50%) | 1 |
| | Positive | 1 (100%) | 0 (0%) | |
| Tissue transglutaminase antibodies | Negative | 9 (50%) | 9 (50%) | **0.04** |
| | Positive | 33 (80%) | 8 (20%) | |

**Note:**
$P < 0.05$, significant.

**Table 3 Correlation between fibromyalgia impact questionnaire and celiac disease parameters.** Correlation values.

| Pathological type of biopsy | Type 1 (4) | Type 2 (3) | Type 3 (5) | P-value |
|---|---|---|---|---|
| Fibromyalgia impact questionnaire | 62.55 ± 9.92 | 73.72 ± 9.09 | 53.22 ± 23.11 | 0.301(a) |
| | 64.71 (48.69–2.11) | 75.94 (63.72–81.49) | 42.38 (32.3–87) | |
| **Anti-gliadin antibodies** | **Negative (12)** | **Positive (4)** | | |
| Fibromyalgia impact questionnaire | 60.1 ± 19.61 | 55.44 ± 11.81 | | 0.664(t) |
| | 64.79 (32.3–87) | 56.12 (42.38–67.12) | | |
| **Clinical type of CD** | **Typical (11)** | **Silent (6)** | | |
| Fibromyalgia impact questionnaire | 56.99 ± 19.4 | 65.34 ± 13.79 | | 0.368(t) |
| | 51.79 (32.3–83.74) | 65.36 (43.65–87) | | |
| **Tissue transglutaminase antibodies** | **Negative (9)** | **Positive (8)** | | |
| Fibromyalgia impact questionnaire | 63.99 ± 18.11 | 55.37 ± 17.09 | | 0.33(t) |
| | 63.72 (35.15–87) | 57.85 (32.3–75.94) | | |
| **Endomysium antibodies** | **Negative (12)** | **Positive (4)** | | |
| Fibromyalgia impact questionnaire | 58.67 ± 19.59 | 62.24 ± 15.12 | | 0.746(t) |
| | 63.64 (32.3–87) | 59.4 (48.69–81.49) | | |

**Note:**
Stats: Mean ± SD/Median (Minimum–Maximum); (a) One-way ANOVA; (s) Student's $t$-test.

## DISCUSSION

The present study evaluated the relationship between CD and FMS. The frequency of fibromyalgia was lower in patients who were negative for tissue transglutaminase

antibodies and endomysium antibodies compared with those who were positive for these antibodies. This appears to suggest that patients who are positive for these antibodies are more likely to have FMS. However, there was no significant relationship between the FIQ, WPI, and SS scales related to the severity of fibromyalgia and celiac parameters.

The question of whether the diagnosis of CD should be included in population-wide screening (or even outside studies) has so far received a negative answer. It is therefore understandable to examine subgroups which can benefit from screening for CD.

CD is a multisystemic autoimmune disorder affecting 1–2% of the population (mostly women), associated with a persistent intolerance to gluten (a type of protein found in bread, pasta, cookies, pizza crust, and many other foods containing wheat, barley, or rye). Most people are carriers of one of two HLA-II genotypes, DQ2 and DQ8 (*Husby et al., 2020*; *Sahin & Mermer, 2022*). Among these diseases, gliadin peptides weaken the immune system, cause the production of tTG autoantibodies and prevent the spread of small intestinal diseases such as villous atrophy, intraepithelial lymphocytosis and crypt hyperplasia. Although symptoms of CD can occur at any age and along with symptoms of bowel and/or bowel diseases, some diagnoses can also be made in asymptomatic people. In particular, a gluten-free diet leads to better healing and successful colonization in the majority of CD patients (*Crowe, 2011*; *Rodrigo, 2006*; *Sahin, 2021*).

In contrast, FMS is a complex disease that affects approximately 2% of the world's population (mostly women) and is characterized by a variety of soft tissue pain, motor problems, abnormal fatigue, non-restorative sleep, and many other symptoms. Its pathogenesis remains unclear, and no analytical tests or imaging techniques are currently available for objective diagnosis. Therefore, a diagnosis of exclusion and inclusion of characteristic features should be used for patients who meet the American College of Rheumatology 1990 criteria for FMS, provided that other types of diseases have been excluded following a thorough evaluation (*Rodrigo, 2006*; *Wolfe et al., 1990*; *Sarzi-Puttini et al., 2020*). Another problem with FMS is the lack of effective treatment to control the symptoms; It causes great suffering for individuals, families and communities and leads to increased use of health services (drinking, absenteeism, disability and early retirement) (*Arnold et al., 2019*).

The surprising results of the current study suggest that the autoimmune inflammatory process associated with gluten in the digestive tract may lead to the development or exacerbation of central nervous system sensitivity, which is very responsible in some individuals with CD or gluten sensitivity (*Yunus, 2012*). In particular, this hypothesis appears consistent with reports of a higher prevalence of FMS in women with various inflammatory diseases of the gastrointestinal tract and in patients with a long history of abdominal pain before the disease. The comorbid triad of CD, fatigue, and musculoskeletal pain has been shown to be particularly difficult, and other authors have suggested that it may indicate a food allergy (*Smith, Harris & Clauw, 2011*; *Zipser et al., 2003*; *Makharia et al., 2022*; *Ahonen et al., 2023*).

The results of this study may be biased due to the small number of patients involved and the lack of long-term follow-up of patients with CD and FMS. However, our results are promising because, in addition to providing new research on CD and FMS, research on

undetected cases of CD-associated FMS will facilitate early initiation and prevention of treatment, adding to the misunderstanding of scientific knowledge. Thus, simultaneous treatment of gastrointestinal and extraintestinal symptoms can provide a benefit that can accomplish two objectives with one action. In the long term, it may also prevent further CD-related complications in patients clearly at risk for CD and in undiagnosed relatives.

## CONCLUSION

Presence of clinical extraintestinal manifestations in patients with CD should lead clinicians to consider FMS as a possible diagnosis. This suggests that physicians of all specialties should be aware of the many symptoms and conditions associated with FMS. This approach can both increase the success of treatment and prevent infections related to FMS treatment. This issue warrants controlled studies with longer follow-up periods.

### Funding
The authors received no funding for this work.

### Competing Interests
The authors declare that they have no competing interests.

### Author Contributions
- Mehmet Serkan Kılıçoğlu conceived and designed the experiments, performed the experiments, prepared figures and/or tables, authored or reviewed drafts of the article, and approved the final draft.
- Safiye Sayılır performed the experiments, authored or reviewed drafts of the article, and approved the final draft.
- Ozan Volkan Yurdakul conceived and designed the experiments, prepared figures and/or tables, authored or reviewed drafts of the article, and approved the final draft.
- Teoman Aydin analyzed the data, prepared figures and/or tables, authored or reviewed drafts of the article, and approved the final draft.
- Koray Koçhan conceived and designed the experiments, prepared figures and/or tables, and approved the final draft.
- Metin Basaranoglu analyzed the data, prepared figures and/or tables, authored or reviewed drafts of the article, and approved the final draft.
- Okan Kucukakkas analyzed the data, prepared figures and/or tables, authored or reviewed drafts of the article, and approved the final draft.

### Human Ethics
The following information was supplied relating to ethical approvals (*i.e.*, approving body and any reference numbers):

Bezmialem Vakif University granted Ethical approval to carry out the study within its facilities (Ethical Application Ref: 2019-3546).
## Data Availability

The raw measurements are available in the Supplemental File.

## Supplemental Information

Supplemental information for this article can be found online at http://dx.doi.org/10.7717/peerj.17949#supplemental-information.

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
