# Peer review of "Association between celiac disease and fibromyalgia and their severity: a cross-sectional study"

_PeerJ, doi:10.7717/peerj.17949_

## Round 0.1 · original submission · Major Revisions

Dear Dr. Kılıçoğlu,

If you feel that you can appropriately revise your paper in response to the reviewers' comments, please submit a revised manuscript. Please also include a response to each of the reviewers' comments, explaining how you have revised the manuscript in response to the reviewers' comments.

Yours,

Yoshi

Prof. Yoshinori Marunaka, M.D., Ph.D.
Academic Editor
PeerJ Life & Environment

Reviewer 1 ·

Basic reporting

Dear Editor,
I should first thank for inviting me as potential reviewer to read and comment on paper entitled ‘‘Association between celiac disease and fibromyalgia and their severity: A cross-sectional 2 study’’.

In the current study, the authors aimed to examine clinical manifestations in adult patients with CD, to investigate the frequency and severity of fibromyalgia that co-occurs with CD, and to analyze the possible relationship between the two diseases.

The main title accurately reflects the major topic and content of the study.
The abstract summarizes and reflects the work described in the manuscript. Also, the abstract presents the significant points related to the background, objectives, materials and methods, results and conclusions.

The materials and methods sufficiently described for the results and conclusions that are presented in the preceding sections. The study type and design were defined in the section of the materials and methods. Inclusion and exclusion criteria are well defined. Tables are sufficient and well-organized. Ethics Committee approval was received. So, the section materials and methods is adequate.

The statistical methods used are appropriate.

The section of the discussion is well organized. The conclusions are drawn appropriately supported by the literature. The manuscript adequately describes the background, present status and significance of the study. The manuscript interprets the findings adequately and appropriately, highlighting the key points clearly.

I think that it will contribute to the literature. I have some minor criticisms.

- According to ESPGHAN updated guidelines, it is not needed to make a definitive diagnosis of celiac disease. Why did the authors perform the colonoscopy to all patients study? Please state and explain it in the manuscript.
- The manuscript appropriately cites the important and authoritative references but does not cite the recent published articles.
- In the manuscript, HLA aassociation of celiac disease is stated but no reference is cited.
…‘’Most people are 254 carriers of one of two HLA-II genotypes, DQ2 and DQ8’’…

Experimental design

Appropriate

Validity of the findings

Appropriate

·

Basic reporting

This is a very interesting and unique study that explores the incidence of fibromyalgia in patients diagnosed with celiac disease. However, there are major flaws that need to be addressed.
1. In the abstract (lines 40-44) should be stated in the positive rather that the negative. Those with antibodies more frequently met criteria for fibromyalgia (P= 0.04, P=0.04 respectively).
2. There is no reference for the ESPGHAN criteria (line 118).
3. It is not clear to me that the 4 clinical types are widely accepted (line 104). Another reference is needed.
4. Are these criteria validated for adults?
5. What was the presenting symptoms? They were seen in Physical Medicine & Rehabilitation (line 115) so it may have selected those with musculoskeletal symptoms rather than a random group of patients with known celiac disease.
6. There are 2 definitions of WPI – a 27 item questionnaire (line 158) and presence of pain in 19 body parts (line 163). The criteria for fibromyalgia uses the latter.
7. Line 221 17 of 60 is 28% (it is correct in the table).
8. The percents could be rounded to whole numbers (easier to read and more accurate given only 60 patients).
9. The discussion could be shortened and more focused.
10. Line 268 – fibromyalgia is a diagnosis of both exclusion and inclusion of characteristic features, just like any other clinical diagnosis. It is not just a diagnosis of exclusion.
11. Lines 290-292 – these is not understandable and not complete sentences. Not sure it is necessary.
12. References. See above and many are not properly formatted.
13. Table 1. The last 3 items are not in English
14. The raw data:
a. Numbers over 19 appear in Widespread pain
b. Numbers over 12 appear in Symptom Severity scale
c. They need a column of meeting criteria for fibromyalgia (WPI ≥ 7 with SSS ≥ 5 or WPI = 3-6 and SSS ≥ 9)

Experimental design

Adequate if 4 classes of CD can be referenced.

Validity of the findings

Needs work - see above. Not sure those with fibromyalgia met criteria.

Additional comments

see above. I liked the idea of this study and I think it is a useful contribution.

---

## Round 0.2 · Minor Revisions

Dear Dr. Kılıçoğlu,

One reviewer has recommended minor revision but could not see the appropriate responses from the authors in the manuscript.

Please resubmit a revised version of your manuscript with a complete response to each of the reviewers' comments.

Yours,

Yoshi

Prof. Yoshinori Marunaka, M.D., Ph.D.

Reviewer 1 ·

Basic reporting

Dear Editor,
I have recommended minor revision but I could not see the appropriate responses from the authors in the manuscript.

If the authors revise the manuscript with the appropriate responses, it would be better and can be acceptable for publication.

Minor criticisms and recommendations:

- According to ESPGHAN updated guidelines, it is not needed to make a definitive diagnosis of celiac disease. Why did the authors perform the colonoscopy to all patients study? Please state and explain it in the manuscript.

According to ESPGHAN updated guidelines, Gastroduodenoscopy should be done, not colonoscopy.
Please explain ‘’Why did the authors perform the colonoscopy to all patients study?’’

- The manuscript appropriately cites the important and authoritative references but does not cite the recent published articles. If the recent published article about celiac disease for example ‘’ Celiac disease in children: A review of the literature. World J Clin Pediatr. 2021 Jul 9;10(4):53-71’’ are cited, the manuscript would be better.

- In the manuscript, HLA aassociation of celiac disease is stated but no reference is cited. The recent published reference about that for example ‘’ Frequency of celiac disease and distribution of HLA-DQ2/DQ8 haplotypes among siblings of children with celiac disease. World J Clin Pediatr 2022;11: 351-359‘’ is cited, the manuscript would be better.
…‘’Most people are 254 carriers of one of two HLA-II genotypes, DQ2 and DQ8’’…

Experimental design

N/A

Validity of the findings

N/A

Additional comments

Dear Editor,
I have recommended minor revision but I could not see the appropriate responses from the authors in the manuscript.

If the authors revise the manuscript with the appropriate responses, it would be better and can be acceptable for publication.

Minor criticisms and recommendations:

- According to ESPGHAN updated guidelines, it is not needed to make a definitive diagnosis of celiac disease. Why did the authors perform the colonoscopy to all patients study? Please state and explain it in the manuscript.

According to ESPGHAN updated guidelines, Gastroduodenoscopy should be done, not colonoscopy.
Please explain ‘’Why did the authors perform the colonoscopy to all patients study?’’

- The manuscript appropriately cites the important and authoritative references but does not cite the recent published articles. If the recent published article about celiac disease for example ‘’ Celiac disease in children: A review of the literature. World J Clin Pediatr. 2021 Jul 9;10(4):53-71’’ are cited, the manuscript would be better.

- In the manuscript, HLA aassociation of celiac disease is stated but no reference is cited. The recent published reference about that for example ‘’ Frequency of celiac disease and distribution of HLA-DQ2/DQ8 haplotypes among siblings of children with celiac disease. World J Clin Pediatr 2022;11: 351-359‘’ is cited, the manuscript would be better.
…‘’Most people are 254 carriers of one of two HLA-II genotypes, DQ2 and DQ8’’…

---

## Round 0.3 · accepted · Accept

Dear Dr. Kılıçoğlu,

Congratulations again, and thank you for your submission.

Warm regards,
Yoshi
Prof. Yoshinori Marunaka, M.D., Ph.D.

Reviewer 1 ·

Basic reporting

Thanks authors for the study. The authors did an appropriate point-by-point response.
I think that it will contribute to the literature.

Experimental design

N/A

Validity of the findings

N/A

Additional comments

None

·

Basic reporting

it is much better.

Experimental design

it is fine.

Validity of the findings

Their finding are not overstated.

Additional comments

They addressed my concerns, although on Table 1, there are still numbers for Symptom Severe Score above 12 and even if you subtract the Mean WPI from the Mean SSS it is above 12.